# LARP7-like protein Pof8 regulates telomerase assembly and poly(A)+TERRA expression in fission yeast

Amanda K. Mennie[1], Bettina A. Moser[1] & Toru M. Nakamura [1]

Telomerase is a reverse transcriptase complex that ensures stable maintenance of linear eukaryotic chromosome ends by overcoming the end replication problem, posed by the inability of replicative DNA polymerases to fully replicate linear DNA. The catalytic subunit TERT must be assembled properly with its telomerase RNA for telomerase to function, and studies in *Tetrahymena* have established that p65, a La-related protein 7 (LARP7) family protein, utilizes its C-terminal xRRM domain to promote assembly of the telomerase ribonucleoprotein (RNP) complex. However, LARP7-dependent telomerase complex assembly has been considered as unique to ciliates that utilize RNA polymerase III to transcribe telomerase RNA. Here we show evidence that fission yeast *Schizosaccharomyces pombe* utilizes the p65-related protein Pof8 and its xRRM domain to promote assembly of RNA polymerase II-encoded telomerase RNA with TERT. Furthermore, we show that Pof8 contributes to repression of the transcription of noncoding RNAs at telomeres.

[1] Department of Biochemistry and Molecular Genetics, College of Medicine, University of Illinois at Chicago, Chicago, IL 60607, USA. Correspondence and requests for materials should be addressed to T.M.N. (email: nakamut@uic.edu)

Telomeres, ends of linear chromosomes, play critical roles in the stable maintenance of eukaryotic genomic DNA, and a number of genetic diseases in patients suffering premature aging involve mutations in proteins that ensure telomere maintenance[1,2]. Regulation of telomere maintenance also affects progression of cancer[3]. In most eukaryotic cells, telomeric DNA consists of repetitive GT-rich sequences, maintained by the specialized reverse transcriptase telomerase, with its tightly associated RNA serving as a template for de novo extension of telomeric repeats[4]. In the absence of this telomerase ribonucleoprotein (RNP) complex, telomeric repeats are progressively lost due to the inability of replicative DNA polymerases to fully replicate ends of linear DNA[5].

Telomeric repeats consist of both double-stranded DNA (dsDNA) and 3′ single-stranded DNA (ssDNA) overhangs, also known as G-tails[6]. Upon loss of telomeric repeats, cells lose their ability to protect telomeres against various DNA damage response and repair proteins due to the loss of telomere-specific protective complexes[7]. In mammalian cells, dsDNA telomeric repeat DNA-binding proteins TRF1 and TRF2, G-tail binding protein Pot1, and additional subunits RAP1, TIN2, and TPP1 assemble into the telomere protective complex shelterin, which plays critical roles not only in regulation of DNA damage responses but also in telomerase regulation[6]. In addition, the CST (CTC1–STN1–TEN1) complex, which also binds to G-tails, plays critical roles in coordinating actions of telomerase and the lagging strand DNA polymerases[8].

The fission yeast Schizosaccharomyces pombe, with well-conserved shelterin and CST complexes, has emerged as an attractive model organism to study telomere regulation[9]. The fission yeast shelterin complex consists of Taz1 (TRF1/TRF2 ortholog), Rap1, Poz1, Tpz1 (TPP1 ortholog), Ccq1, Pot1, and Rif1[10–12]. While Taz1, Rap1, Poz1, Rap1, and Rif1 are required to limit telomere extension by telomerase, Tpz1 and Ccq1 are required for recruitment and activation of telomerase[10–16]. Interaction between the telomerase regulatory subunit Est1 and Ccq1 is required to recruit telomerase to telomeres, and this interaction is promoted by Rad3[ATR]/Tel1[ATM]-dependent phosphorylation of Ccq1 Thr93[17]. Taz1, Rap1, and Poz1 are necessary to limit Ccq1 Thr93 phosphorylation, while Tpz1-Ccq1 interaction is essential for Thr93 phosphorylation and optimal localization of Ccq1 at telomeres[15,17,18]. In addition, Tpz1-Ccq1 interaction plays a critical role in enforcing transcriptional repression at telomeric and sub-telomeric regions by promoting telomere localization of the chromatin regulatory complexes SHREC (Snf2/HDAC-containing repressor complex) and CLRC (Clr4 histone H3K9 methyltransferase complex)[15,19–21].

A systematic telomere length analysis of fission yeast gene deletion strains has identified a large number of mutants that influence telomere length[22]. As expected, mutants with very short or no telomere included deletion mutants for the telomerase catalytic subunit Trt1 (TERT), its regulatory subunit Est1, and the shelterin subunit Ccq1. Interestingly, deletion of the putative F-box protein Pof8, which interacts with the SCF ubiquitin ligase complex subunit Skp1[23,24] but had no previously known telomere function, was identified to cause very short telomeres.

In our current study, we investigated how Pof8 contributes to telomere maintenance in fission yeast, and found that Pof8 tightly associates with telomerase RNA, ensures telomerase RNA accumulation in cells, and promotes assembly of telomerase RNA with Trt1. In addition, we found that Pof8 represses expression of long noncoding RNAs (lncRNAs)[25,26] at telomeric and sub-telomeric regions. Sequence analysis indicated that Pof8 is likely an ortholog of telomerase subunits from ciliate protozoans, Tetrahymena thermophila p65 and Euplotes aediculatus p43, which belong to the LARP7 (La-related protein 7) family[27–29] and are

known to play a critical role in telomerase RNP assembly[30,31]. While LARP7 family proteins in other organisms have been well established to play a critical role in regulation of RNA polymerase II (Pol II)-dependent transcription[29,32] as a component of the 7SK RNA-containing RNP, it is currently believed that regulation of telomerase RNP assembly is a unique feature of ciliate LARP7 proteins[29,33]. Furthermore, proteins unrelated to LARP7 have been implicated in assembly of mammalian or budding yeast telomerase RNP, leading to the suggestion that 'species-specific' telomerase assembly factors are utilized in telomerase RNP formation[33–35]. Thus, our current finding that a conserved p65/p43-related protein promotes telomerase assembly in fission yeast challenges this long held belief, and suggests that LARP7-dependent telomerase RNP assembly might represent an ancient and well conserved mechanism.

## Results

**Pof8 promotes telomere elongation by telomerase.** Confirming a previous systematic telomere length analysis[22], we found that pof8Δ cells maintain substantially shorter but stable telomeres (Fig. 1a). Telomere maintenance in pof8Δ cells remained dependent on telomerase, since pof8Δ trt1Δ cells completely lost telomeric repeats upon restreaks, much like trt1Δ cells[13]. On the other hand, the homologous recombination repair protein Rad52 was not required to maintain telomeres in pof8Δ cells (Fig. 1a). While removal of the shelterin components Taz1, Rap1, Poz1, and Rif1 led to substantial telomerase-dependent telomere elongation[10–13], elimination of Pof8 reversed telomere elongation observed in these mutants (Fig. 1b), further supporting the notion that Pof8 plays an important role in telomerase-dependent telomere elongation.

Chromatin Immunoprecipitation (ChIP) assays revealed that telomere association of both telomerase subunits Trt1 and Est1 is reduced in pof8Δ cells, although not as severely as in a deletion mutant for telomerase RNA (ter1Δ; Fig. 1c, d). We also found reduced expression of Trt1, but not Est1, in pof8Δ and ter1Δ cells, with more severe reduction in ter1Δ (Fig. 1c, d). Nevertheless, telomerase is still localized to telomeres, consistent with the finding that telomerase is required for telomere maintenance in pof8Δ cells.

ChIP assays further revealed that Pof8 is recruited to telomeres, and that its telomere association is substantially reduced but not completely eliminated in deletion mutants for telomerase subunits (ter1Δ, trt1Δ, and est1Δ) or in ccq1Δ cells[12,14] (Fig. 1e). We also found that Pof8 shows similar binding at non-telomeric sites in ter1[+] and ter1Δ backgrounds, while Trt1 is not recruited to any of the non-telomeric genomic loci tested (Supplementary Fig. 1). Thus, Pof8 appears to be broadly associated with various chromosome regions, independently of telomerase.

**Pof8 tightly binds TER1 and promotes Trt1 binding to TER1.** While Pof8 was previously reported as a putative F-box protein that interacts with Skp1[23,24], bioinformatic analysis of Pof8 from four Schizosaccharomyces species and Pof8-like proteins from additional fungal species found that Pof8 is likely to share a similar overall domain organization to the well-established ciliate telomerase subunits p65 from Tetrahymena[28] and p43 from Euplotes[27], which are members of the LARP7 family proteins[29,32] (Fig. 2, Supplementary Figs. 2, 3 and Supplementary Table 5). Mammalian LARP7 protein binds constitutively to an abundant RNA Polymerase III (Pol III) transcript, the non-coding 7SK RNA, and has been established as a regulator of the transcription elongation factor P-TEFb for RNA Pol II[32,36]. LARP7 family proteins contain three major conserved sequence motifs: the adjacent Lupus antigen (La)-motif and RNA recognition motif

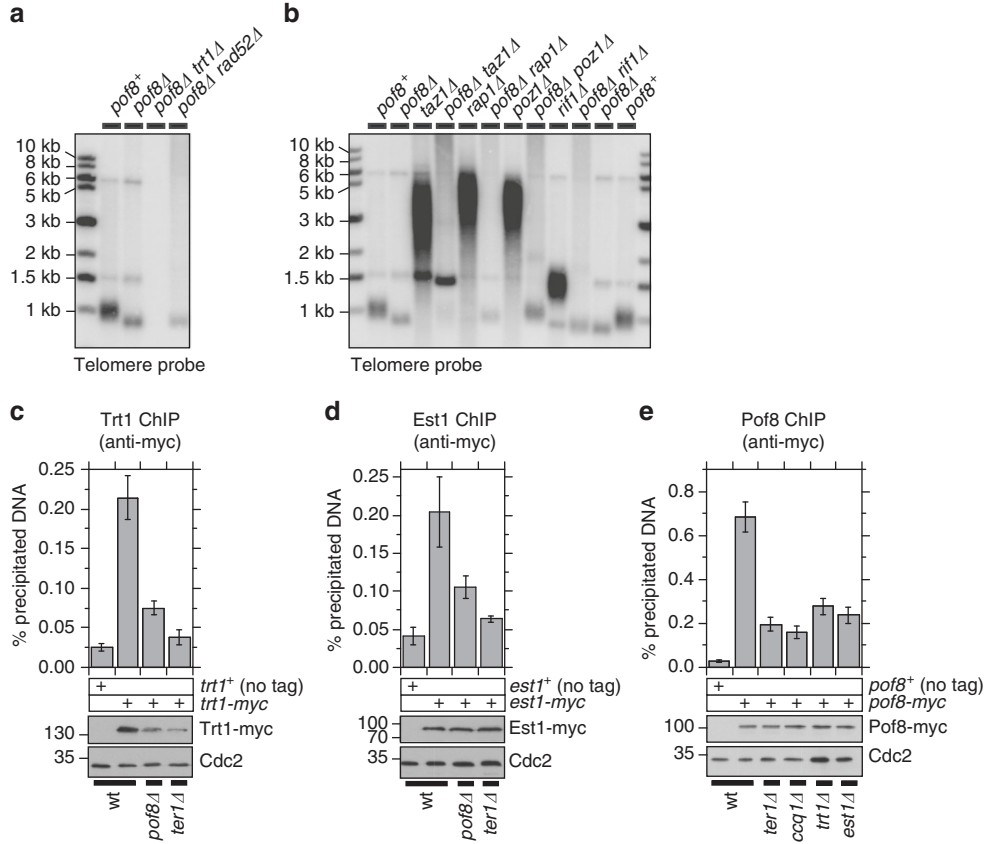

**Fig. 1** Pof8 contributes to telomerase-dependent telomere maintenance. **a**, **b** Southern blot analysis of indicated strains probed with *S. pombe* telomere repeats. After strains were generated by transformation or genetic crosses, they were extensively restreaked on YES plates to achieve terminal telomere length. **c–e** Quantitative real-time PCR ChIP analysis to monitor association with telomeres for (**c**) Trt1, (**d**) Est1, and (**e**) Pof8 for indicated genetic backgrounds. Plots show mean values plus/minus SEM from at least 5 independent experiments. Raw data and statistical analysis are available in Supplementary Data 1. Expression levels of myc-tagged proteins used in ChIP assays were monitored by western blot analysis. Anti-Cdc2 blots served as loading control. Molecular weight (kDa) of size markers are also indicated

(RRM) at the N-terminus which together constitute a La-module, and a C-terminal xRRM motif[29,32,33] (Fig. 2a). The La-module is sufficient to recognize and bind the 3′ poly(U) tract of RNA Pol III transcripts, while the xRRM domain is not required for this function[32,33]. The xRRM domain of *Tetrahymena* p65 interacts with the Stem Loop IV of telomerase RNA TER and induces a conformational change in telomerase RNA, which in turn facilitates the assembly of the telomerase catalytic subunit TERT with TER RNA[37]. Similarly, the xRRM domain of mammalian LARP7 protein contributes to the assembly of the 7SK RNP complex[29,32,33].

The C-terminal xRRM in Pof8 showed the most convincing sequence conservation with LARP7 family proteins (Fig. 2b), and this region has already been assigned as RRM domain in the NCBI Conserved Domain Database (https://www.ncbi.nlm.nih.gov/cdd) for fission yeast Pof8 and related proteins from other fungal species (Supplementary Fig. 2). By contrast, the region with putative La-motif homology, especially at its N-terminal half did not appear to be well conserved (Supplementary Fig. 3a). However, the C-terminal region of the putative La-motif and its closely associated RRM region (RRM1) showed better conservation with other LARP7 family members (Supplementary Fig. 3a). In addition, immediately following the N-terminal RRM, fungal Pof8 proteins showed an additional homology region (designated as RRM1-C) that is predicted to form a β-sheet followed by an α-helix, based on structure-based alignment analysis by PROMAL3D[38]. Interestingly, this region also showed homology to a previously identified C-terminal region of LARP7 that shows

homology to the α3 helix region of RRM1 in genuine La proteins[39] (Fig. 2a and Supplementary Fig. 3b). Taken together, we thus considered that Pof8 might indeed be closely related to the LARP7 family proteins.

To test whether Pof8 might fulfill an analogous function as the ciliate p65 and p43 proteins in assembly of the telomerase RNP complex, we examined whether it can stably associate with the telomerase RNA TER1. Remarkably, we detected very robust co-immunoprecipitation (co-IP) between Pof8 and TER1 (Fig. 3a), comparable or even better than co-IP between the telomerase catalytic subunit Trt1 and TER1 (Fig. 3b). Furthermore, *pof8Δ* cells showed substantially reduced interaction between Trt1 and TER1 (Fig. 3b). Conversely, deletion of Trt1 substantially reduced the TER1-Pof8 interaction (Fig. 3a). Thus, Pof8, and Trt1 appear to mutually promote interaction with TER1 RNA to allow efficient assembly of the TER1-Trt1-Pof8 complex, analogous to the telomerase 'catalytic core' complex of TER-TERT-p65 found in *Tetrahymena* cells[31,35,37]. Consistently, we found that Trt1 can indeed co-IP with Pof8 efficiently (Fig. 3d). Trt1-Pof8 interaction was sensitive to RNase A treatment, suggesting that TER1 RNA mediates the assembly of the TER1-Trt1-Pof8 complex.

Interestingly, deletion of Ccq1 also substantially reduced the TER1-Pof8 interaction (Fig. 3a). Since Ccq1 is also required for efficient localization of Pof8 at telomeres (Fig. 1e), Ccq1 has emerged as a very important regulator of Pof8 function at telomeres. On the other hand, deletion of Est1 did not affect TER1-Pof8 interaction (Fig. 3a), and deletion of Pof8 in turn did not affect TER1-Est1 interaction (Fig. 3c). Thus, Est1 interaction

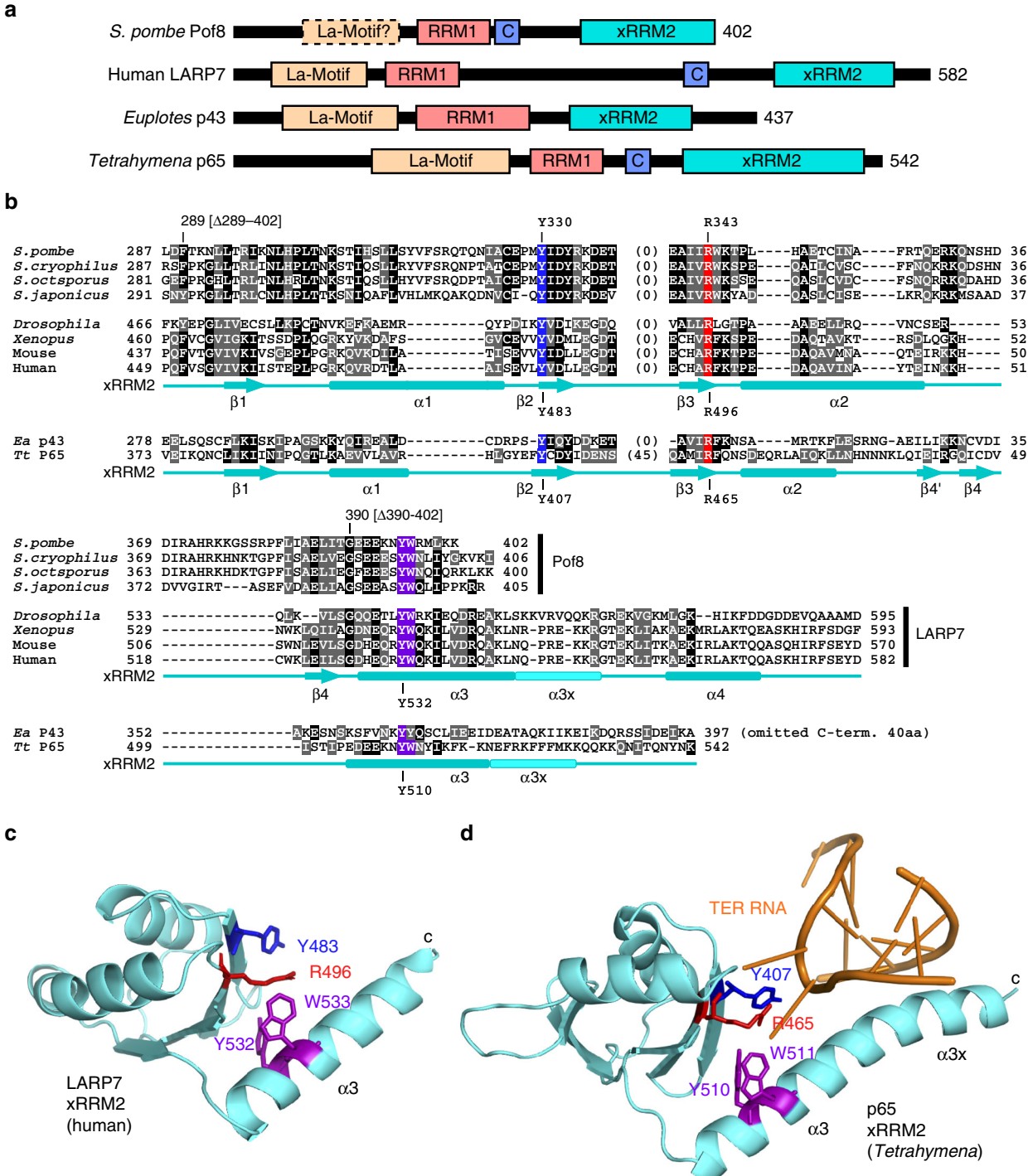

**Fig. 2** Fission yeast Pof8 shows similar domain organization as LARP7 family proteins. **a** Comparison of putative functional domains identified in *S. pombe* Pof8 to corresponding domains in human LARP7, *Euplotes* p43 and *Tetrahymena* p65 proteins. **b** Sequence alignment of xRRM2 domain from Pof8 proteins from *Schizosacchromyces* species (*S. pombe*, *S. cryophilus*, *S. octsporus*, and *S. japonicus*), LARP7 proteins (*Drosophila*, *Xenopus*, mouse, and human), *Euplotes* p43, and *Tetrahymena* p65. Highly conserved residues that have been implicated in RNA recognition[33, 37, 47] are highlighted with colored background. Amino-acid residues that show at least 50% conservation among aligned sequences are highlighted in black (identical residues) or gray (similar residues, grouped as GAVLI, FYW, CM, ST, KRH, DENQ, and P). The secondary structures for human LARP7[47] and *Tetrahymena* p65[37] are also indicated in cyan. Full length alignment of Pof8 from four fission yeast species, along with Pof8-like proteins from additional fungal species are shown in Supplementary Fig. 2. Sequence alignment of the N-terminal domain spanning the putative La-motif and RRM1 domains are shown in Supplementary Fig. 3a. **c**, **d** Structures of xRRM2 domains from (**c**) human LARP7[47] and (**d**) *Tetrahymena thermophila* p65[37]. Highly conserved amino acid residues are shown with same color as highlighted in (**b**). For LARP7, amino acid residues V455-N545 are shown (PDB ID: 5KNW). For p65, amino-acid residues C379-N412 and G461-K532 are shown (PDB ID: 4ERD). (Extra amino acid residues between N412 and G461 were deleted in the construct that was used in structure determination of p65-TER complex[37].) Telomerase RNA shown in **d** corresponds to G118-U127 and A144-A148

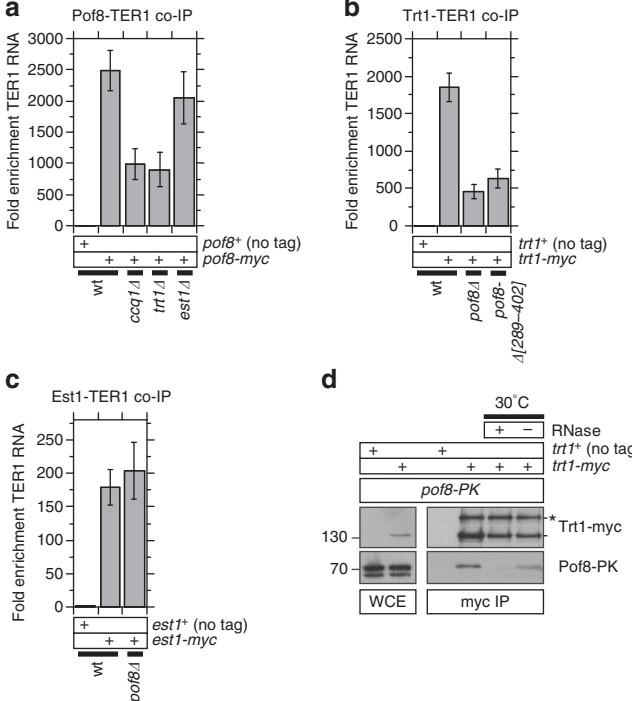

**Fig. 3** Pof8 binds strongly to telomerase RNA and facilitates interaction between Trt1 and telomerase RNA. **a** Binding of Pof8 to telomerase RNA TER1 was reduced in *ccq1Δ* and *trt1Δ*, but not in *est1Δ* cells. Error bars correspond to SEM from at least 8 independent experiments. **b** TER1-Trt1 interaction was greatly diminished but not completely eliminated in *pof8Δ* or *pof8-Δ[289–402]* cells. Error bars correspond to SEM from at least 4 independent experiments. **c** TER1-Est1 interaction was not affected by elimination of Pof8. Error bars correspond to SEM from at least 6 independent experiments. All TER1 co-IP plots show fold enrichment of TER1 RNA recovered after IP compared to no tag control strain. Raw data and statistical analysis are available in Supplementary Data 1. **d** Examination of Trt1-Pof8 interaction by co-IP. The Trt1-Pof8 interaction was nearly eliminated upon 10 min incubation at 30 ºC in the presence of 1 mg/mL RNase A, while mostly retained in mock treatment without RNase A, suggesting that Trt1-Pof8 interaction is mediated by TER1 RNA. As previously reported[70], anti-myc pull down of Trt1-myc showed enrichment of a higher molecular weight band marked with asterisk (*) not easily observed in whole cell extract (WCE). Molecular weight (kDa) of size markers are also indicated

with the rest of the telomerase RNP complex appears to be independent of Pof8, and that Est1 is not required for Pof8's ability to promote TER1-Trt1 interaction.

**Pof8 promotes Lsm binding to TER1 and protects TER1.** LARP7 family proteins play a critical role in protecting their tightly associated RNA against nucleolytic degradation[29,32]. Since ciliate telomerase RNA or mammalian 7SK RNA are transcribed by RNA Pol III, they end with a run of Uracil (U) residues, which is recognized by the N-terminal La-module of p65 and LARP7 proteins[31,39]. Fission yeast telomerase RNA is transcribed by RNA Pol II and initially carries a poly(A)-tail[40]. It also carries a consensus Sm binding site $AU_{4-6}G$ and an intron near its 3' end. The Sm complex promotes a spliceosome-dependent 'slicing' reaction, wherein the RNA undergoes only the initial cleavage step of the splicing reaction, without completing the ligation step that allows joining of exons[41,42]. This initial slicing reaction is crucial for the further maturation of the telomerase RNP

complex, which includes the addition of a 5' tri-methyl-G cap to telomerase RNA, exchange of the Sm complex with the Sm-like complex Lsm2-8, and assembly of Trt1 with telomerase RNA[42]. The spliceosome-mediated cleavage also truncates one nucleotide from the Sm binding site of telomerase RNA, which is further subjected to exonucleolytic degradation so that the majority of mature telomerase RNA ends with a string of U residues[42], a preferred binding site for LARP7 family proteins and perhaps Pof8. On the other hand, 3' poly(U) is also a preferred binding site of the Lsm2-8 complex, and mutation of the poly(U) sequence disrupts both TER1-Lsm and TER1-Trt1 interactions[42].

Based on these previous findings, we wondered if Pof8 might protect telomerase RNA against degradation by collaborating with the Lsm2-8 complex, and we indeed found that *pof8Δ* cells show substantial reduction in cellular TER1 RNA levels (Fig. 4a). By contrast, TER1 RNA levels were not significantly affected in *ccq1Δ*, *trt1Δ* or *est1Δ* cells (Supplementary Fig. 4). Furthermore, we observed no change in the level of the precursor form of TER1 RNA in *pof8Δ* cells (Fig. 4b). Thus, while Pof8 is important for maintaining mature TER1 levels, it appears to function downstream of the spliceosome-mediated maturation of TER1.

In addition, we found that interaction between the Lsm2-8 complex subunit Lsm3 and TER1 is eliminated in *pof8Δ* cells (Fig. 4c). Thus, the reduction in TER1 RNA level could be caused at least in part by failure to properly assemble the Lsm2-8 complex at the 3' end of TER1. However, since TER1 co-IP efficiency for Lsm3 is substantially lower than that of Pof8 (Figs. 3a and 4c), it seems likely that not all Pof8-associated TER1 is also associated with the Lsm2-8 complex. In fact, while TER1 RNA-dependent Pof8-Lsm3 interaction (Fig. 4d) and Pof8-dependent Trt1-Lsm3 interaction (Fig. 4e) could be detected by co-IP, these interactions were harder to detect than the robust Trt1-Pof8 interaction (Fig. 3d). Thus, the reduction in TER1 RNA levels in *pof8Δ* might not entirely be attributable to loss of interaction with the Lsm complex. Nevertheless, results from current and previous experiments[40,42–45] are consistent with the existence of a telomerase holoenzyme complex in fission yeast consisting of TER1, Trt1, Pof8, Est1, and the Lsm2-8 complex, with interactions among protein subunits mediated by TER1 RNA (Fig. 4g).

Interestingly, ChIP assays revealed that Lsm3 is recruited to fission yeast telomeres, but its association with telomeres was independent of Pof8, TER1, Trt1 or Est1 (Fig. 4f). Thus, while the Lsm2-8 complex associates with a catalytically active telomerase RNP complex[42], localization of the Lsm complex itself does not depend upon its ability to associate with telomerase RNA, or any of the other telomerase subunits. In fact, *ter1Δ* cells surprisingly showed a substantial increase in telomere association of Lsm3 (Fig. 4f), indicating that the presence of TER1 RNA might actually limit the accumulation of the Lsm complex at telomeres. We also found that Lsm3 can bind to non-telomeric sites, that is significantly increased in *ter1Δ* cells (Supplementary Fig. 1). Thus, it appears that TER1 RNA also plays a totally unexpected global role in limiting accumulation of the Lsm complex to various chromosomal regions.

**Restored TER1 level fails to rescue *pof8Δ* telomere defects.** The fact that *pof8Δ* cells show substantial reduction in TER1 RNA level raised the question as to whether reduction in TER1 alone is sufficient to explain telomere and telomerase dysfunctions associated with *pof8Δ*. To address this possibility, we expressed TER1 RNA from an exogenous expression plasmid under the control of an inducible mid-level *nmt1* promoter[46] to test if forced expression of TER1 can reverse telomere defects observed in *pof8Δ* cells.

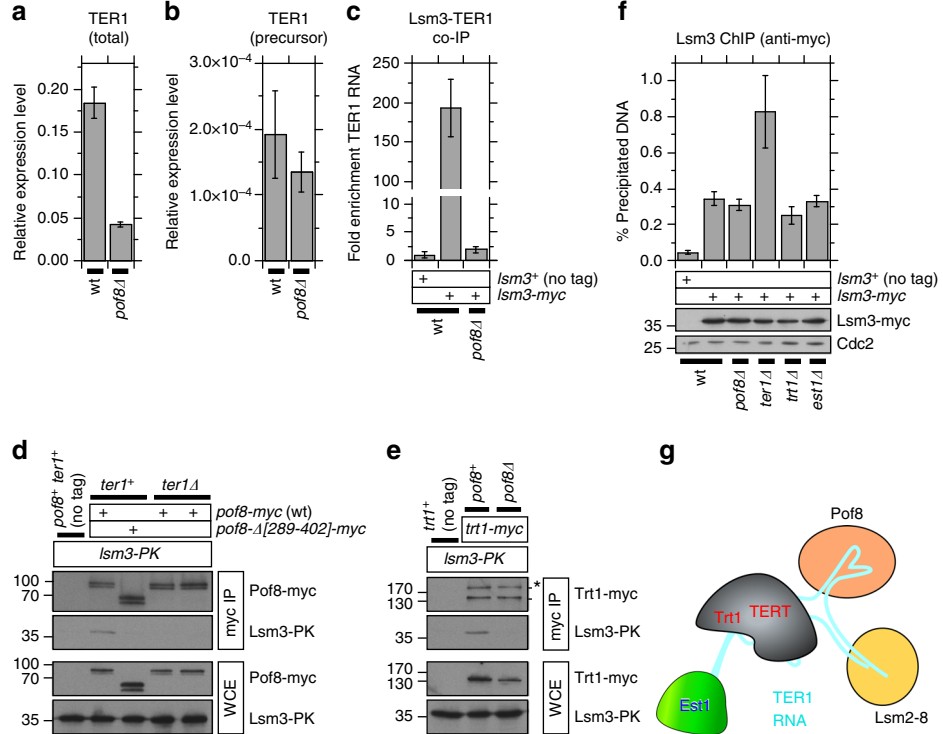

**Fig. 4** Pof8 is important for maintaining cellular levels of TER1 RNA. **a** Elimination of Pof8 leads to ~4.3-fold reduction in cellular telomerase RNA level. **b** Expression level of TER1 precursor RNA, still carrying an intron near the 3′ end of RNA, was not reduced in pof8Δ cells. For **a** and **b**, TER1 RNA expression levels were normalized to his1⁺ mRNA expression. **c** TER1-Lsm3 interaction was eliminated in pof8Δ. The plot shows fold enrichment of TER1 RNA recovered after IP compared to no tag control strain. **d**, **e** Examination of (**d**) Lsm3-Pof8 and (**e**) Lsm3-Trt1 interactions by co-IP. Trt1 band marked with asterisk (*) in myc-IP blot corresponds to previously reported[70] higher molecular weight band. **f** Quantitative real-time PCR ChIP assays showed that telomere association of Lsm3 is independent of Pof8 or the telomerase subunits TER1, Trt1, and Est1. Unexpectedly, Lsm3 binding was increased when telomerase RNA was eliminated. Expression levels of myc-tagged Lsm3 used in ChIP assays were monitored by western blot analysis. Anti-Cdc2 blots served as loading control. Molecular weight (kDa) of size markers are also indicated. Error bars for plots in **a** through **c** and **f** correspond to SEM from at least 6 independent experiments. Raw data and statistical analysis are available in Supplementary Data 1. **g** A model of fission yeast telomerase holoenzyme complex based on current and previous co-IP data

While introduction of the TER1 expression plasmid restored cellular TER1 RNA levels in pof8Δ cells to levels higher than observed in pof8⁺ cells (~2-fold when repressed in the presence of thiamine B1, and ~3-fold when induced in the absence of thiamine B1; Fig. 5a), it failed to restore telomere length in pof8Δ cells (Fig. 5b). Conversely, introduction of the TER1 expression plasmid fully restored telomere length in ter1Δ cells, confirming that TER1 RNA expressed from the nmt1 promoter is functional (Fig. 5b). Furthermore, over-expression of TER1 only weakly restored TER1-Trt1 interaction (Fig. 5c) and failed to rescue Trt1 recruitment at telomeres (Fig. 5d) in pof8Δ cells. Taken together, these results thus established that Pof8 plays a critical role in TER1-Trt1 assembly and telomere maintenance that is separable from its role in maintaining TER1 RNA expression.

**Pof8 xRRM domain is required for Pof8 binding to TER1.** The C-terminal xRRM domain of Pof8 showed highest conservation with other LARP7 family proteins (Fig. 2). To understand the functional significance of the xRRM domain, we generated point mutants pof8-Y330A and pof8-R343A, as well as C-terminal truncation mutants pof8-Δ[390–402] and pof8-Δ[289–402] (Fig. 2b). Y330 and R343 were mutated since these highly conserved residues have been identified as crucial points of direct RNA interaction in Tetrahymena p65 and human LARP7[37,47] (Fig. 2c, d). The Δ[390–402] truncation specifically removed the putative α3 helix, that is critical for RNA chaperon activity in p65

and LARP7 proteins, while the Δ[289-402] truncation removed the entire xRRM domain.

Southern blot analysis revealed that all four xRRM domain mutants lead to a similar extent of telomere shortening as pof8Δ (Fig. 6a). While western blot analysis of myc-tagged Pof8 mutants indicated that all four xRRM domain mutants were stably expressed in vivo at levels comparable to wild-type Pof8 (Fig. 6d), they failed to efficiently interact with TER1, establishing that the xRRM domain is essential for TER1-Pof8 interaction (Fig. 6b). We also found that all xRRM domain mutants showed significantly reduced cellular TER1 RNA levels comparable to pof8Δ cells (Fig. 6c). In addition, pof8-Δ[289–402] caused reduced TER1-Trt1 co-IP comparable to pof8Δ (Fig. 3b), and eliminated Pof8-Lsm3 interaction detected by co-IP (Fig. 4d), further establishing the importance of the Pof8 xRRM domain in assembling a functional telomerase RNP complex.

Intriguingly, even though these Pof8 mutants failed to interact with TER1, they still robustly bound to telomeres. In fact, three out of four xRRM domain mutants showed telomere association on par or even greater than wild-type Pof8, except Δ[390–402] showing a partial reduction in binding (Fig. 6d). Furthermore, ChIP analysis revealed that xRRM-domain mutants show only modest or no reduction of Trt1 binding to telomeres (Fig. 6e). These data thus indicated that while the xRRM domain is critical for TER1-Pof8 interaction and accumulation of mature TER1, it is surprisingly not strictly required to promote telomere association of Trt1. On the other hand, since even xRRM mutants that show

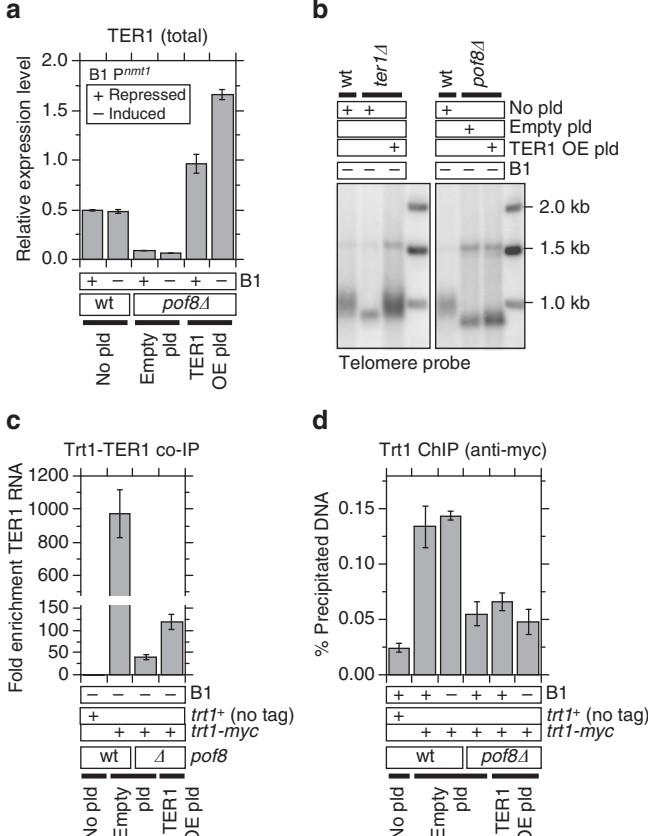

**Fig. 5** Over-expression of TER1 is unable to suppress telomerase defect of *pof8Δ* cells. **a** Introduction of a *nmt1* promoter-controlled TER1 expression plasmid allowed restoration of TER1 RNA levels in *pof8Δ* cells to ~2-fold or ~3-fold over *pof8+* wild-type cells when grown in minimum media under repressed (+B1) or induced (−B1) conditions, respectively. TER1 RNA expression levels were normalized to *his1+* mRNA expression. Error bars correspond to SEM from 3 independent experiments. **b** Southern blot analysis of telomere length for indicated strains. While TER1 plasmid fully restored wild-type telomere length in *ter1Δ* cells, it was unable to suppress telomere shortening in *pof8Δ* cells. (**c**) Restoration of TER1 RNA expression failed to restore TER1-Trt1 interaction in *pof8Δ* cells. Plot shows fold enrichment of TER1 RNA recovered after IP compared to no tag control strain. **d** Restoration of TER1 RNA expression failed to restore Trt1 recruitment at telomeres. Error bars for plots in **c** and **d** correspond to SEM from at least 4 independent experiments. Raw data and statistical analysis are available in Supplementary Data 1

robust Trt1 recruitment cannot maintain wild-type telomere length (Fig. 6a), the xRRM domain is required for Pof8's ability to promote telomerase-dependent telomere maintenance.

**xRRM is dispensable for repression of telomere transcripts.** Based on ChIP analysis, we found that *pof8Δ* cells show reduced Ccq1 binding at telomeres (Fig. 7a). Thus, while Ccq1 protein promotes TER1-Pof8 interaction (Fig. 3a) as well as telomere localization of Pof8 (Fig. 1e), Pof8 in turn appeared to contribute to Ccq1 function at telomeres. Ccq1 is required not only for telomerase recruitment but also for transcriptional inhibition at telomeric and sub-telomeric regions in fission yeast[15,19–21]. Therefore, we next examined whether transcriptional silencing at telomeres is affected in *pof8Δ* cells by utilizing a reporter strain that carries a *ura4+* gene ~ 300 bp away from the left telomere of chromosome II[48]. Consistent with the notion that *pof8Δ* is defective in transcriptional repression at telomeres, we observed

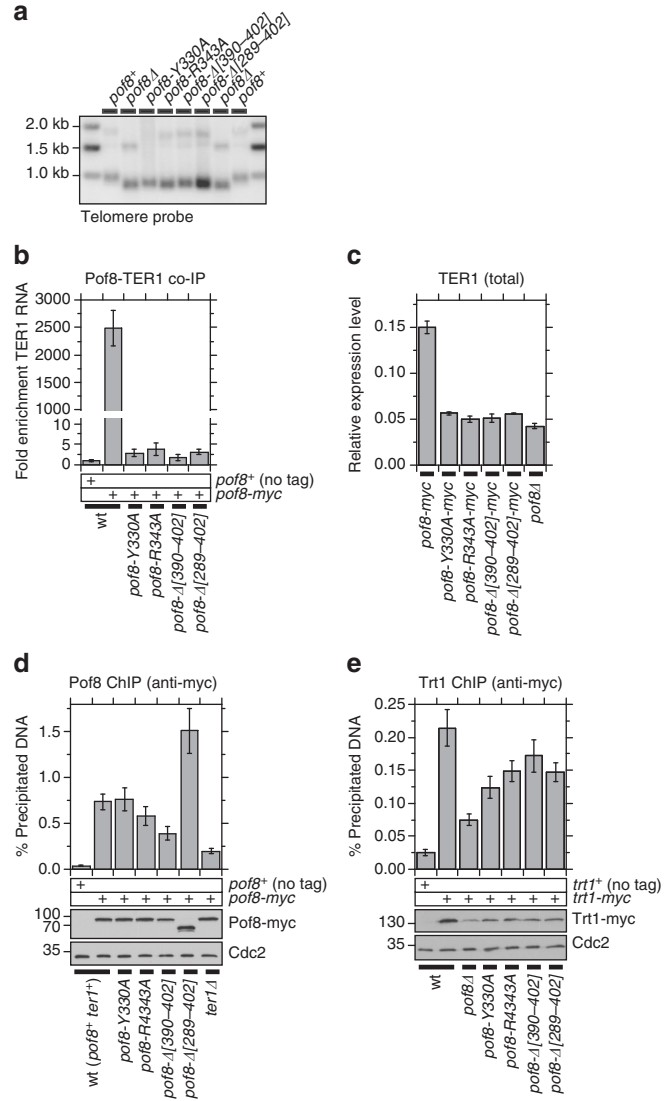

**Fig. 6** xRRM domain of Pof8 is important for TER1-Pof8 interaction and accumulation of TER1 RNA. **a** Southern blot analysis showed that xRRM domain mutants show telomere shortening comparable to *pof8Δ*. Genomic DNA samples from indicated strains were prepared after extensive restreaks on YES plates prior to analysis to achieve terminal telomere length. **b** TER1-Pof8 interaction was essentially eliminated in xRRM domain mutants of Pof8. The plot shows fold enrichment of TER1 RNA recovered after IP compared to no tag control strain. Error bars correspond to SEM from at least 6 independent experiments. **c** The xRRM domain mutants failed to accumulate telomerase RNA. TER1 RNA expression levels were normalized to *his1+* mRNA expression. Error bars correspond to SEM from at least 5 independent experiments. **d**, **e** Quantitative real-time PCR ChIP analysis to monitor association with telomeres for (**d**) Pof8 and (**e**) Trt1 in indicated genetic backgrounds. Error bars correspond to SEM from at least 12 (**d**) and 8 (**e**) independent experiments. Raw data and statistical analysis are available in Supplementary Data 1. Expression levels of (**d**) Pof8 and (**e**) Trt1 used in ChIP assays were monitored by western blot analysis. Anti-Cdc2 blots served as loading control. Molecular weight (kDa) of size markers are also indicated

that *pof8Δ* cells, carrying the telomeric *ura4+* reporter, were able to grow on minimal media lacking uracil, much like *ccq1Δ* cells (Fig. 7b).

Previous studies found that fission yeast telomere/sub-telomere regions express complex sets of RNA Pol II transcribed long

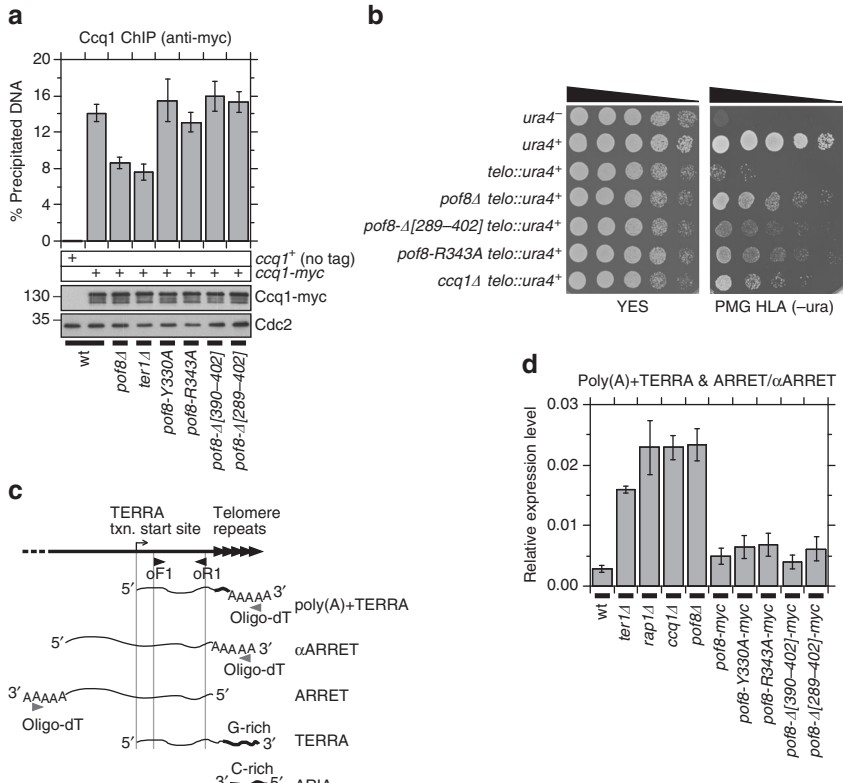

**Fig. 7** Pof8 protein is required to repress transcription at telomere/sub-telomere regions. **a** Quantitative real-time PCR ChIP analysis to monitor Ccq1 association with telomeres for indicated genetic backgrounds. Expression levels of myc-tagged Ccq1 used in ChIP assays were monitored by western blot analysis. Anti-Cdc2 blots served as loading control. Molecular weight (kDa) of size markers are also indicated. **b** Transcriptional silencing at telomeres is disrupted in *pof8Δ* cells. Wild-type and *pof8* mutant cells carrying the *ura4⁺* marker gene at the chromosome 2 L telomere[48] were serially diluted and spotted on YES (no selection) or PMG HLA (-ura) plates. **c** A schematic diagram indicating various lncRNA species expressed at fission yeast telomere/sub-telomere regions[26, 49]. Locations of primers used in RT reaction (oligo-dT) and subsequent quantitative PCR analysis (oF1 and oR1) to monitor poly (A)-tailed lncRNAs are indicated. **d** Pof8 protein, but not its xRRM domain is required to repress transcription of poly(A)-tailed lncRNAs at telomere/sub-telomere regions. Expression levels of telomeric transcripts were normalized to *his1⁺* mRNA. For plots in **a** and **d**, error bars correspond to SEM from at least five independent experiments. Raw data and statistical analysis are available in Supplementary Data 1

noncoding RNA (lncRNA) species with distinct structures, termed TERRA, poly(A)+TERRA, ARIA, ARRET, and αARRET[25,26,49] (Fig. 7c). Both TERRA and poly(A)+TERRA RNAs are transcribed from a TERRA transcription start site within the sub-telomeric region towards telomeric ends. While TERRA RNA contains a G-rich strand of telomeric repeat sequence and associates with telomeres, poly(A)+TERRA contains only a few telomeric repeats and terminates with a poly(A)-tail. ARIA RNA is essentially made up of a C-rich strand of telomeric repeats, while ARRRET and αARRET RNAs contain complementary strands of subtelomeric sequence and terminate with a poly(A) tail. Interestingly, poly(A)+TERRA RNA co-IP with telomerase and show increased expression upon telomere shortening caused by *ter1Δ* mutation[49], suggesting that it might be involved in telomerase regulation. Previous studies have also reported that *rap1Δ* cells show an increase in RNA Pol II binding at the telomere/sub-telomere region, as well as an increase in expression of all telomere/sub-telomere-associated lncRNA species[26,50].

To test whether Pof8 is involved in repression of native lncRNAs, we isolated RNA from wild-type and *pof8Δ* cells, and adapted a RT-PCR-based assay established by the Azzalin lab[26,49] to monitor poly(A)-tailed lncRNAs species (poly(A)+TERRA, ARRRET, and αARRET; Fig. 7c). As controls for strains that show increased expression of telomeric transcripts, we also included *ter1Δ*, *rap1Δ* and *ccq1Δ* cells in our analysis.

In our RT-PCR assay, we detected a substantial increase in expression of RNA in *pof8Δ* cells, consistent with the notion that Pof8 contributes to repression of telomeric lncRNAs (Fig. 7d). This increase in expression appears to be mainly attributable to poly(A)+TERRA, since a RT-PCR assay designed to specifically detect ARRET and αARRET showed more modest increase in expression in *pof8Δ* cells (Supplementary Fig. 5). The extent of increase was greater than that observed in *ter1Δ* cells and was comparable to that of *rap1Δ* and *ccq1Δ* cells (Fig. 7d). Interestingly, pof8 xRRM domain mutants did not show significant reduction in Ccq1 binding at telomeres (Fig. 7a). In addition, *pof8-Δ[289–402]* and *pof8-R343A* cells showed milder de-repression of telomeric *ura4⁺* reporter than *pof8Δ* cells (Fig. 7b), and all four pof8 xRRM domain mutants did not induce expression of the telomeric transcripts to the same extent as *pof8Δ* cells (Fig. 7d). Thus, it appears that the N-terminal regions of Pof8 that encompass the putative divergent La-module can contribute not only to the telomere association of Pof8 (Fig. 6d) and Trt1 (Fig. 6e), but also to Ccq1-dependent repression of telomere transcription (Fig. 7d), independently of the C-terminal xRRM domain.

## Discussion

The current study uncovered an unexpected conservation of function by the putative LARP7-like protein Pof8 in fission yeast

telomerase biogenesis. While *Tetrahymena* p65 and *Euplotes* p43 proteins from ciliated protozoa are well-established members of the LARP7 family that function to promote proper assembly of the telomerase RNA with the catalytic subunit TERT[30,31,37], this has long been thought to be a unique feature of the ciliate telomerase complex[29,33,35]. Thus, it was unexpected and exciting that fission yeast Pof8 is likely to be an ortholog of p65/p43 that also functions in assembly of the holoenzyme telomerase RNP complex consisting of TER1, Trt1, Pof8, Lsm2-8, and Est1 (Fig. 4g).

Since 'Pof8-like' proteins across diverse fungal species share an overall domain organization and display good sequence conservation with *S. pombe* Pof8 (Supplementary Fig. 2), these proteins could have a conserved role in telomerase assembly. Previous studies have uncovered that (1) spliceosome-dependent cleavage of telomerase RNA precursor by an aborted splicing reaction that carries out only the 1st step of the splicing reaction, (2) the existence of a poly(U) sequence near the 3′ end of the precursor exon that can potentially serve as binding site for Sm and Lsm complexes, and (3) a three-way junction (TWJ) structure that resembles the CR4/5 region of vertebrate telomerase RNA, are conserved features among telomerase RNAs from diverse fungal species[51–54]. Considering that LARP7 family members function as RNA chaperons that protect their tightly bound RNAs with 3′ end poly(U) tract against degradation while promoting RNP assembly[29,32,33], conservation of the 3′ poly(U) sequence might be important not only for Sm/Lsm binding but also for the ability of Pof8/LARP7-like proteins to promote assembly of telomerase RNP in fungal species. On the other hand, divergence in the La-motif sequence could have been driven by the requirement of Pof8-like proteins not to interfere with the poly(U) recognition by Sm/Lsm complexes in telomerase biogenesis. In fact, several residues within the La-motif that were previously identified as playing critical roles in recognition of the 3′ poly(U) tail are not conserved in fungal Pof8-like proteins[32,39] (Supplementary Figs. 2, 3). The finding that Pof8 is essential for the Lsm complex to interact with TER1 RNA (Fig. 4c) is also consistent with the notion that Pof8 in fission yeast (and perhaps in other fungal species) has evolved to collaborate with the Lsm complex in telomerase RNA protection and telomerase RNP assembly.

Previous studies found that the TWJ structure at the CR4/5 domain near the 3′ end of vertebrate telomerase RNA binds tightly to the Telomerase RNA Binding Domain (TRBD) of TERT, and undergoes substantial conformational change upon binding to TRBD[55]. Therefore, the CR4/5 domain has been proposed to carry out the equivalent function to the Loop IV region of *Tetrahymena* telomerase RNA, which must undergo a p65 xRRM-domain dependent conformational change to interact with *Tetrahymena* TERT[35,54,55]. The TWJ structure of fission yeast telomerase RNA binds tightly to the TRBD of Trt1, and plays a critical role in telomerase activity[55,56]. Thus, the Pof8 xRRM domain, much like the xRRM domain of *Tetrahymena* p65[31,35,37], might induce a conformational change of the TWJ to facilitate TER1-Trt1 assembly. Since Trt1 promotes TER1-Pof8 interaction (Fig. 3a), it is possible that Trt1 and Pof8 closely collaborate in the conformational change of the TWJ to promote productive assembly of the fission yeast telomerase core RNP complex TER1-Trt1-Pof8. By contrast, we found that the TER1-Est1 interaction, which appears to be much weaker and/or occurs in a sub-stoichiometric manner compared to robust TER1-Trt1 or TER1-Pof8 interaction, is not dependent on Pof8 (Fig. 3c). Est1 association with the core telomerase RNP complex may be more dynamically regulated in order to allow telomerase to transition from an initial recruitment phase, involving Ccq1-Est1 interaction, to an 'activation' phase to allow telomerase to extend telomeres[16,17,57].

Characterization of Lsm3, a subunit of the Lsm complex, has provided new and unexpected insights. Eukaryotic cells contain distinct Lsm complexes: the Lsm2-8 complex that functions in nuclear snRNP assembly, the cytoplasmic Lsm1-7 complex that functions in regulation of mRNA turnover, and the Lsm2-7 complex that regulates RNase P RNA maturation in the nucleolus[58,59]. The Lsm2-8 complex, but not Lsm1-7, specifically associates with the catalytically active telomerase RNP complex in fission yeast[42]. Thus, we predicted that the Lsm complex is localized to telomeres in a telomerase-dependent manner. However, we found that while the Lsm complex subunit Lsm3 is indeed localized to telomeres, its binding to telomeres hardly changed in *trt1Δ* or *est1Δ* cells, and it unexpectedly showed elevated binding in *ter1Δ* cells (Fig. 4f). Furthermore, while *pof8Δ* disrupted TER1-Lsm3 interaction (Fig. 4c), it had no effect on the ability of Lsm3 to be localized to telomeres (Fig. 4f). Thus, the Lsm complex appears to be an integral part of fission yeast telomeres independent of telomerase. Since Lsm3 is shared among different Lsm complexes, it is currently unknown if all or subsets of these complexes are recruited to telomeres. While the Lsm1-7 complex is expected to function primarily in the cytoplasm to regulate mRNA decay, a previous study found that Lsm1 is localized to both, the nucleus and cytoplasm[60]. Considering that lncRNA TERRA associates with telomeric DNA[26], Lsm3 might be recruited to telomeres by telomere-associated TERRA as part of either Lsm1-7, Lsm2-8, or Lsm2-7 complex. On the other hand, since we also detected binding of Lsm3 at various non-telomeric sites and binding of Lsm3 was substantially increased at all telomeric and non-telomeric sites we examined upon elimination of TER1 RNA (Fig. 4f and Supplementary Fig. 1), it seems likely that TER1 RNA plays a totally unexpected regulatory role in limiting chromosome association of the Lsm complex at many loci throughout the fission yeast genome.

LARP7 has been extensively studied as an integral part of the 7SK RNA-containing RNP, which serves an evolutionarily conserved function in regulation of RNA Pol II-dependent transcription from invertebrates to humans[36,47]. While it remains unclear if Pof8 might interact with additional RNAs besides telomerase RNA, or if Pof8 might have non-telomeric roles in regulation of transcription, we did find increased expression of telomere/sub-telomere encoded RNA Pol II-dependent lncRNAs in *pof8Δ* cells (Fig. 7d). This effect could be telomere specific, since Pof8 association with telomeres is partially dependent on telomerase, and *pof8Δ* also reduced telomere association of Ccq1, a protein previously found to interact with heterochromatin regulators SHREC and CLRC[15,19–21]. On the other hand, since we detected telomerase-independent binding of Pof8 at the telomere and several non-telomeric loci (Fig. 1e and Supplementary Fig. 1), it would be interesting to investigate if Pof8 might fulfill telomerase-independent non-telomeric function(s).

It was recently reported that knock down of LARP7 reduces telomerase activity in HeLa cells, and that Alazami syndrome patients carrying frame-shift mutations in LARP7 show a premature telomere shortening phenotype[61]. Such findings, along with our current report, might thus suggest that the LARP7 family proteins share a conserved role as an RNA chaperon in the assembly of an active telomerase complex. Although previous studies did not identify LARP7 as a potential mammalian telomerase subunit[34,62], it is possible that mammalian LARP7 interacts with telomerase RNA only transiently and thus was lost during purification. Alternatively, genuine La protein, rather than LARP7, might fulfill mammalian telomerase RNP assembly, since it does interact with the mammalian telomerase complex[34,63], and it shares a similar domain organization as LARP7, including the C-terminal xRRM domain[33]. Thus, regardless of the possibility, the xRRM domain could represent a highly conserved functional

domain critical for telomerase RNP assembly. Since the xRRM domain of Pof8 is much better conserved than its putative La-motif, it is tempting to suggest that the defining feature of a possible 'universal' RNA chaperon for telomerase RNP assembly is not the La-motif but rather the xRRM motif.

Finally, while our current investigation focused on similarities between fission yeast Pof8 and the ciliate LARP7 family members p65 and p43, it is also important to note that fission yeast Pof8 was identified as a putative F-box protein, and has been experimentally shown to interact with Skp1 protein, but surprisingly not with the Cullin protein Cul1, an integral component of the SCF (Skp-Cullin-F-box) complex[23,24]. Intriguingly, biochemical purification of the *Tetrahymena* telomerase complex identified p65, the CST complex (p75-p45-p19) and Skp1 (p20), but not Cullin, to co-purify with the telomerase catalytic subunit TERT[64]. Thus, it is possible that Skp1, in association with Pof8/p65, might provide the conserved function in telomerase regulation, independent of its more widely studied role in SCF-dependent regulation of ubiquitination.

Taken together, our current investigation has shown that fission yeast Pof8, a protein with a well-conserved xRRM domain, is likely to fulfill similar roles as LARP7 family proteins from ciliates in telomerase regulation. Further investigations of fission yeast Pof8 function will likely provide novel insights into how Pof8- and LARP7-related xRRM domain containing proteins contribute to telomere regulation in a wide variety of organisms including humans.

## Methods

**Yeast strains and molecular biology reagents**. Fission yeast strains used in this study were constructed and cultured using standard methods[65]. For most experiments, cells are grown in YES (Yeast Extract with Supplements) media at 32 °C, while they are grown in PMG (Pombe Minimal Glutamate) with appropriate supplements to allow for selection of either *LEU2* or *ura4*+ plasmid at 32 °C for experiments involving TER1 expression plasmids[65]. Fission yeast strains used in this study are listed in Supplementary Table 1. Original sources of published strains and details on strain construction for newly generated strains are indicated in Supplementary Table 2. Plasmids used in this study are listed in Supplementary Table 3. Primers used in this study are listed in Supplementary Table 4. Various point mutation and truncation constructs were generated by Q5 (NEB, E0554) or QuikChange Lightning (Agilent, 210513) site-directed mutagenesis.

**Southern blot analysis**. Genomic DNA was prepared by resuspending cell pellets in equal volumes of DNA buffer (100 mM Tris pH 8.0, 1 mM EDTA pH 8.0, 100 mM NaCl, 1% SDS) and Phenol:chloroform:isoamyl alcohol (25:24:1) saturated with 10 mM Tris, pH 8.0 and 1 mM EDTA, and cells were lysed with glass beads. Subsequently, 800 ng genomic DNA was digested overnight with EcoRI, separated on a 1% agarose gel, transferred to Amersham Hybond-XL (GE Healthcare), hybridized to $^{32}$P labeled telomeric DNA probe, and quantified with GE Typhoon Phosphorimager[15,66].

**Immunoprecipitation and western blot analysis**. Cells were resuspended in lysis buffer (50 mM Tris pH 8.0, 150 mM NaCl, 10% glycerol, 5 mM EDTA pH 8.0, 0.5% NP-40, 50 mM NaF, 1 mM DTT, 1 mM Na₃VO₄, 1 mM PMSF, Roche complete protease inhibitor cocktail), and lysed with glass beads[66]. Proteins were immunoprecipitated using monoclonal anti-myc antibody (9B11, Cell Signaling) and Dynabeads Protein G (Thermo Fisher). Western blot analyses were performed using monoclonal anti-myc (9B11) at 1:5000, monoclonal anti-V5/PK (SV5-Pk1, BioRad) at 1:1000, or monoclonal anti-Cdc2 (y100.4, Abcam) at 1:10000 dilutions as primary antibodies. HRP-conjugated (goat) anti-mouse (Pierce, 31430) was used as the secondary antibody at 1:5000 dilution. Uncropped scans of all western blot figures are shown in Supplementary Fig. 6.

**ChIP assay**. Cells were crosslinked with formaldehyde by addition of 1/10 volume of fixation solution (11% formaldehyde, 100 mM NaCl, 1 mM EDTA pH 8.0, 0.5 mM EGTA pH 8.0, 100 mM Tris-HCl pH 8.0) for 20 min at room temperature, incubated additional 5 min after addition of Glycine at final concentration of 125 mM, washed 3× with TBS (20 mM Tris-HCl pH 7.6, 150 mM NaCl), pelleted, and frozen in liquid nitrogen[15,67]. Cell pellets were resuspended in lysis buffer (50 mM Hepes-KOK pH 7.5, 140 mM NaCl, 1 mM EDTA, 1% Triton X-100, 0.1% SDS, Roche complete protease inhibitor cocktail, 1 mM PMSF), processed for ChIP[15,67] using monoclonal anti-myc (9B11), and analyzed with quantitative real-time PCR with primers listed in Supplementary Table 4. ChIP sample values were normalized

to Input samples and plotted as % precipitated DNA. Raw data values of individual experiments and statistical analysis are available in Supplementary Data 1.

**TER1 immunoprecipitation assay**. Cells were lysed with glass beads in TMG buffer (10 mM Tris pH 8.0, 1 mM MgCl₂, 100 mM NaCl, 1 mM EDTA pH 8.0, 10% glycerol, 1 mM DTT, 1 mM PMSF, 0.5% Tween-20, Roche complete protease inhibitor cocktail), and processed for telomerase co-IP with monoclonal anti-myc (9B11)[44,66]. RNA was purified and treated with rDNase using NucleoSpin RNA columns (Macherey-Nagel, 740955) and reverse transcribed using M-MLV Reverse Transcriptase (Promega, M1705) with Primer 1016, and subsequently analyzed by quantitative real-time PCR analysis with primers 1015 and 1017 (Supplementary Table 4). Control reactions were also performed without reverse transcriptase to ensure that the PCR signal reflected RNA and not contaminating DNA. % IP efficiencies of TER1 RNA over Input TER1 RNA for individual strains were first determined, and these values were then normalized to the average % IP for no tag control samples, and plotted as fold-enrichment for TER1 RNA recovered after IP compared to no tag control. Raw data values of individual experiments and statistical analysis are available in Supplementary Data 1.

**RNA expression level**. Cells were suspended in equal volumes of TES buffer (10 mM Tris pH 7.5, 10 mM EDTA pH 8.0, 0.5% SDS) and acid-equilibrated phenol for 1 h at 65 °C, and processed for RNA extraction[68]. RNA was further purified and treated with rDNase using NucleoSpin RNA columns (Macherey-Nagel). RNA concentration was determined using a Nanodrop 2000 spectrophotometer, and 4 μg was reverse transcribed using M-MLV Reverse Transcriptase (Promega) and subsequently subjected to triplicate SYBR Green-based real-time PCR analysis. Control reactions were also performed without reverse transcriptase to ensure that the PCR signal reflected RNA and not contaminating DNA. Primers used for reverse transcription and real-time PCR are listed in Supplementary Table 4. Expression levels of TER1 and telomeric non-coding RNAs were normalized to expression of *his1*+ mRNA level. Raw data values of individual experiments and statistical analysis are available in Supplementary Data 1.

**Statistical analysis**. Individual raw data values, and statistical analysis for ChIP, TER1 co-IP, and RNA expression level experiments are provided as a Microsoft Excel file in Supplementary Data 1. In addition, graphs for these data values, plotted as either bar graphs or individual points, along with mean values plus/minus standard errors of mean (SEM), are included in Supplementary Data 1. Pairwise two-tailed student's *t*-test and one-way ANOVA analysis were performed with Excel. In addition, post-hoc analysis to control for false discovery rate via the two-stage step-up method of Benjamini, Krieger and Yekutieli[69] was performed with GraphPad Prism software. Significant values ($p < 0.05$) are marked with red letters.

**Data availability**. The authors declare that the data supporting the findings of this study are available within the paper and its Supplementary Information files or upon reasonable request.

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

## Acknowledgements

We thank R.C. Allshire, F. Ishikawa, J. P. Cooper, M. R. Flory, and V. A. Zakian for sharing published strains and plasmids. A.K.M. was supported in part by a UIC Center for Clinical and Translational Science (CCTS) Predoctoral Education for Clinical and Translational Scientists (PECTS) fellowship. We also thank P. Baumann and K. Tomita for communicating their data prior to publication. This work was supported by NIH grant GM078253 (T.M.N.).

## Author contributions

A.K.M. designed, performed and analyzed most of the experiments in this study, and wrote the paper. B.A.M. contributed in generation and characterization of Pof8-related strains, carried out co-IP experiments, and edited the paper. T.M.N. conceived the study, designed experiments, analyzed data, generated fission yeast strains, and wrote the paper.

## Additional information

**Competing interests:** The authors declare no competing financial interests.

