## [Peer Review File · Nature Communications]

Reviewers' comments:

Reviewer #1 (Remarks to the Author):

The paper by Mennie et al describes the role of Pof8 as a critical factor in telomere maintenance. The authors demonstrate that Pof8 is important for stabilization of the telomerase RNA and the assembly of active telomerase. The authors also provide evidence that Pof8 is recruited to telomeres in a telomerase independent pathway that involves ccq1. Moreover, Pof8 deletion elicits another telomere phenotypes, the derepression of TERRA. The paper is well written and experimentally executed. However, due the many phenotypes that are associated with the loss of Pof8, some clarification and additional experiments are needed. Nevertheless, the authors unambiguously demonstrate that Pof8 is a major and previously underappreciated factor in telomere biology. The paper is well suited for publication in Nature communications.

- A short introduction into ccq1 function would be helpful to the reader.

- In Figure 1e, the authors claim telomerase independent recruitment of Pof8 to telomeres. Is this a telomere specific recruitment or does Pof8 localize to other chromatin regions? The same question should be addressed for the LSM3 localization. Does LSM3 localize specifically to telomeres? in a telomerase independent manner.

-Page 12 line 242-247 and Figure 5: the authors suggests that Pof8 also promotes telomerase activity at telomeres. Here, again it is essential to show that the recruitment to chromatin of Pof8 and the mutants is telomere specific and not mere unspecific chromatin association. Also, the authors arguments are not clear or clearly stated. The authors need to provide data that shows that telomerase activities of the RRM mutants is not affected, yet telomeres are short. It seems quite possible that the reduces TER1 levels and the reduced activity causes the telomere length changes.

It would be helpful if the authors could more directly address why telomeres in the Pof8 deletes are short. How do the different functions that the authors uncover contribute? Is TER1 limiting? Is active telomerase limiting? Do the changes in TERRA matter? Along the same line, I find it quite distracting that the reduces levels of TER1 and reduced levels of active telomerase in the Pof8 deletes —which I think is one of the key phenotypes to explain the short telomere phenotype – is not discussed as part of figure 1.

Also, it would be helpful if the authors could more clearly explain how they think about where Pof8 executes it functions. In the currently form the paper could imply that Pof8 is involved in the assembly of mature telomerase while bound to telomeres. Is this what the authors want to say?

Minor concerns:

- The quantification of TER1 level after co-IP should be described better.

- considering the differences in telomere length of the mutants tested, should the TERRA expression levels be normalized to telomere length?

Reviewer #2 (Remarks to the Author):

In the manuscript "LARP7-like protein Pof8 regulates telomerase assembly and poly(A)+TERRA expression in fission yeast," Mennie and colleagues investigate the contribution of the Pof8 protein to telomerase function and telomeric-associated transcription. In ciliates, the LARP7-like proteins p65 (*Tetrahymena thermophila*) and p43 (*Euplotes aediculatus*) are known to associate with the telomerase RNA and induce a conformational change that facilitates telomerase assembly and

stabilizes the RNA against degradation. However, since the telomerase RNA is an RNA pol III transcript, it has been thought that these proteins might have ciliate-specific functions. In contrast, the authors convincingly argue that Pof8, in addition to having sequence similarity with the LARP7-like protein family, associates with the fission yeast telomerase RNA and affects RNA levels. The authors also demonstrate that Pof8 represses transcription at telomeres, both of an ectopic gene placed in the vicinity of a telomere and of endogenous non-coding transcripts. The manuscript is clearly written and adds to our understanding of the complexity of telomerase RNA processing and telomerase assembly. In addition, there are several surprising observations (for example, the finding that Lsm3, a component of the Lsm complex, associates with telomeric DNA even in the absence of its binding partner, the telomerase RNA). There are some concerns as outlined below that can be addressed with additional experiments and statistical analysis of the data.

Major comments:

1. Although the authors indicate in the legends how many times each experiment was repeated and graph the average and standard error of the mean, there is no indication of statistical significance. Such analysis should be done, including adjustment for multiple comparisons when appropriate.
2. The authors conduct a number of ChIP experiments in which the association of a particular protein with telomeric DNA is compared across multiple mutant strain backgrounds. The authors never address the possibility that changes in the % of precipitated DNA reflect differences in protein expression levels in mutant strains. Western blots showing equivalent expression in extract and/or immunoprecipitation during the ChIP procedure should be shown (perhaps in a supplementary figure).
3. I'm concerned whether the reduced TER1 RNA levels in the *pof8* deletion strain contribute to the reduced ability of Trt1 to immunoprecipitate the telomerase RNA. Is "fold enrichment" expressed relative to the untagged control strain, or relative to a measurement of input? Does ectopic expression of TER1 increase RNA levels to those observed in WT? If TER1 RNA levels are restored, is the efficiency of the Trt1 co-IP restored as well? Likewise, are the TER1 RNA levels affected in *ccq1*, *trt1* or *est1* deletion strains (Fig. 3a)?
4. The authors use the reduction of Ccq1 association with telomeres in the *pof8* deletion strain (Fig. 6a) to justify experiments examining telomeric gene silencing (Fig. 6b). It is puzzling, then, that the authors do not include a *ccq1* deletion strain in the transcriptional silencing assays, nor in the assays measuring non-coding RNA abundance (e.g. TERRA). Do the Pof8 mutants that affect RNA abundance and association of Pof8 with the TER1 RNA also affect the association of Ccq1 with telomeres?
5. The authors state that "... Pof8 and Trt1TERT appear to mutually promote interaction with TER1 RNA to allow more efficient assembly of the TER1-Trt1TERT-Pof8 complex, analogous to the telomerase "catalytic core" complex of TER-TERT-p65 found in Tetrahymena cells." However, I don't think that any of the experiments shown prove that Pof8 is found in a ternary complex with TER1 and Trt1. Does immunoprecipitation of Pof8 pull down catalytically active telomerase?
6. The authors state that deletion of *pof8* "completely reversed" (line 101) the telomere elongation observed in mutant strains. However, this doesn't appear to be true for the Taz1 mutant. What is the discrete band at about 1.5 kb that increases in intensity upon Pof8 deletion? Is that pattern reproducible?

Minor comments:

Figure 5b lacks an untagged negative control lane.

Sentence on lines 83 to 85 requires additional references.

Reviewer #1 (Remarks to the Author):

The paper by Mennie et al describes the role of Pof8 as a critical factor in telomere maintenance. The authors demonstrate that Pof8 is important for stabilization of the telomerase RNA and the assembly of active telomerase. The authors also provide evidence that Pof8 is recruited to telomeres in a telomerase independent pathway that involves ccq1. Moreover, Pof8 deletion elicits another telomere phenotypes, the derepression of TERRA. The paper is well written and experimentally executed. However, due the many phenotypes that are associated with the loss of Pof8, some clarification and additional experiments are needed. Nevertheless, the authors unambiguously demonstrate that Pof8 is a major and previously underappreciated factor in telomere biology. The paper is well suited for publication in Nature communications.

We thank Reviewer #1 for a positive overall assessment of our manuscript. As detailed below, we have carefully considered and addressed all points raised by Reviewer #1. I hope that the revised manuscript would be now satisfactory to Reviewer #1. In the revised submission, changes are highlighted with red letters.

- A short introduction into ccq1 function would be helpful to the reader.

In the Introduction section, we have added the following statement.

Tpz1-Ccq1 interaction also plays a critical role in enforcing transcriptional repression at telomeric and sub-telomeric regions in fission yeast by promoting telomere localization of the chromatin regulatory complexes SHREC (Snf2/HDAC-containing repressor complex) and CLRC (Clr4 histone H3K9 methyltransferase complex).

- In Figure 1e, the authors claim telomerase independent recruitment of Pof8 to telomeres. Is this a telomere specific recruitment or does Pof8 localize to other chromatin regions? The same question should be addressed for the LSM3 localization. Does LSM3 localize specifically to telomeres in a telomerase independent manner?

As suggested by Reviewer #1, we performed ChIP assays of several non-telomeric sites for Pof8 and Lsm3 (Supplementary Fig. 1). As a negative control, we have also evaluated Trt1 recruitment at the same non-telomeric loci. Our new results found that both Pof8 and Lsm3 show binding at these non-telomeric sites, but not Trt1.

Perhaps not surprisingly (as telomerase is not detected at these sites), Pof8 binding to these non-telomeric sites did not show any significant difference between wt and *ter1Δ* cells, unlike Pof8 binding at telomeres, which was substantially reduced in *ter1Δ* cells. By contrast, Lsm3 binding showed substantial increase in *ter1Δ* over wt cells at telomeric as well as non-telomeric sites. Thus, it seems that TER1 RNA appears to play an unexpected universal role in restricting association of the Lsm complex at a wide-variety of chromosomal regions.

- Page 12 line 242-247 and Figure 5: the authors suggests that Pof8 also promotes telomerase activity at telomeres. Here, again it is essential to show that the recruitment to chromatin of Pof8 and the mutants is telomere specific and not mere unspecific chromatin association.

Also, the authors arguments are not clear or clearly stated. The authors need to provide data that shows that telomerase activities of the RRM mutants is not affected, yet telomeres are short. It seems quite possible that the reduces TER1 levels and the reduced activity causes the telomere length changes.

It was not our intension to claim that xRRM mutants do not affect telomerase activity, and we apologize for the confusing statement in the original manuscript. In fact, we agree with Reviewer #1 that xRRM mutants will likely have much less active telomerase, especially since our new data found that *pof8-Δ[289-402]* mutant cells also show reduced interaction between Trt1 and TER1 (Fig. 3b). We have revised the relevant section of the manuscript, and we hope it is now clearer why we think that xRRM domain mutants show telomere shortening.

It would be helpful if the authors could more directly address why telomeres in the Pof8 deletes are short. How do the different functions that the authors uncover contribute? Is TER1 limiting? Is active telomerase limiting? Do the changes in TERRA matter? Along the same line, I find it quite distracting that the reduces levels of TER1 and reduced levels of active telomerase in the Pof8 deletes —which I think is one of the key phenotypes to explain the short telomere phenotype – is not discussed as part of figure 1.

By utilizing a TER1 over-expression plasmid, we were able to restore (and in fact express more than wild-type level) TER1 RNA in *pof8Δ* cells (Fig. 5a). However, restoration of TER1 RNA did not rescue telomere shortening or reduced Trt1 recruitment observed in *pof8Δ* cells (Fig. 5b and 5d), and only weakly restored Trt1-TER1 interaction (Fig. 5c).

Thus, while Pof8 is critical for TER1 accumulation, Pof8 additionally contributes to assembly and recruitment of the active Trt1-TER1 complex at telomeres to ensure telomere maintenance.

As for functional significance of induced TERRA expression in *pof8Δ*, we are currently not in a position to address this issue as we do not have a way to reduce TERRA expression in *pof8Δ* cells.

We have also seriously considered moving TER1 expression data to Figure 1, as suggested by Reviewer #1. However, we have ultimately decided to keep the overall structure of the paper, as we wanted to discuss Pof8 sequence analysis and its role in promoting Trt1-TER1 interaction first, and also wanted to provide sufficient background information to explain why we examined both mature and precursor forms of TER1 in *pof8Δ* cells. We did create a new Figure 5, which discusses the TER1 over-expression data.

Also, it would be helpful if the authors could more clearly explain how they think about where Pof8 executes its functions. In the currently form the paper could imply that Pof8 is involved in the assembly of mature telomerase while bound to telomeres. Is this what the authors want to say?

We do not believe that telomerase is assembled on telomeres since we have previously shown that the *ccq1-T93A* mutant, which fails to recruit telomerase to telomeres, did not show any change in Trt1-TER1 association. We have revised our Discussion section to make our model clearer.

Minor concerns:

- The quantification of TER1 level after co-IP should be described better.

We have added additional description to more clearly describe how TER1 co-IP data is calculated in the Method section.

- considering the differences in telomere length of the mutants tested, should the TERRA expression levels be normalized to telomere length?

We feel that the suggested correction is not necessary as we are mostly measuring expression of TERRA-polyA, and this RNA contains only limited telomeric repeat sequence, based on previous reports from the Azzalin lab. It should also be noted that *pof8Δ* and various xRRM domain mutants show identical telomere shortening phenotype, but only *pof8Δ* cells show a large increase in TERRA-polyA RNA expression comparable to *ccq1Δ* and *rap1Δ* cells. Thus, even without any correction, our conclusion that the N-terminal domain of Pof8 contributes to transcriptional repression at telomeres independently of the xRRM domain is fully supported based on the presented data.

Reviewer #2 (Remarks to the Author):

*In the manuscript "LARP7-like protein Pof8 regulates telomerase assembly and poly(A)+TERRA expression in fission yeast," Mennie and colleagues investigate the contribution of the Pof8 protein to telomerase function and telomeric-associated transcription. In ciliates, the LARP7-like proteins p65 (*Tetrahymena thermophila*) and p43 (*Euplotes aediculatus*) are known to associate with the telomerase RNA and induce a conformational change that facilitates telomerase assembly and stabilizes the RNA against degradation. However, since the telomerase RNA is an RNA pol III transcript, it has been thought that these proteins might have ciliate-specific functions. In contrast, the authors convincingly argue that Pof8, in addition to having sequence similarity with the LARP7-like protein family, associates with the fission yeast telomerase RNA and affects RNA levels. The authors also demonstrate that Pof8 represses transcription at telomeres, both of an ectopic gene placed in the vicinity of a telomere and of endogenous non-coding transcripts. The manuscript is clearly written and adds to our understanding of the complexity of telomerase RNA processing and telomerase assembly. In addition, there are several surprising observations (for example, the finding that Lsm3, a component of the Lsm complex, associates with telomeric DNA even in the absence of its binding partner, the telomerase RNA). There are some concerns as outlined below that can be addressed with additional experiments and statistical analysis of the data.*

We also thank Reviewer #2 for the positive overall assessment of our manuscript. As detailed below, we have revised our manuscript and carried out requested experiments. We feel that the additional experiments have substantially strengthened the paper, and thus we hope that it is now satisfactory to Reviewer #2. In the revised submission, changes are highlighted with red letters.

1. Although the authors indicate in the legends how many times each experiment was repeated and graph the average and standard error of the mean, there is no indication of statistical significance. Such analysis should be done, including adjustment for multiple comparisons when appropriate.

In the revised manuscript, we have now included supplemental Excel sheets, where we provide full statistical analysis of all graphs we included in our manuscript, along with individual raw data values.

2. The authors conduct a number of ChIP experiments in which the association of a particular protein with telomeric DNA is compared across multiple mutant strain backgrounds. The authors never address the possibility that changes in the % of precipitated DNA reflect differences in protein expression levels in mutant strains. Western blots showing equivalent expression in extract and/or immunoprecipitation during the ChIP procedure should be shown (perhaps in a supplementary figure).

As requested, we have performed western blots to evaluate protein expression levels in various mutant backgrounds.

3. I'm concerned whether the reduced TER1 RNA levels in the pof8 deletion strain contribute to the reduced ability of Trt1 to immunoprecipitate the telomerase RNA. Is "fold enrichment" expressed relative to the untagged control strain, or relative to a measurement of input?

%IP of TER1 relative to Input TER1 was calculated for a given strain, including no tag control strain. In the final graphs, we represented fold-enrichment in efficiency of TER1 co-IP over no tag control. We have included Excel sheets of raw %IP data in the supplementary material before it was normalized to no tag control. We have also included a description how it was calculated in the method section, and we hope that the additional information will clarify how we calculated our data.

Does ectopic expression of TER1 increase RNA levels to those observed in WT? If TER1 RNA levels are restored, is the efficiency of the Trt1 co-IP restored as well?

We were able to restore TER1 expression level in *pof8* Δ by utilizing a TER1 over-expression plasmid (Fig. 5a). However, Trt1-TER1 co-IP efficiency was only weakly restored (Fig. 5c), and more importantly, TER1 over-expression did not restore telomerase recruitment (Fig. 5d) or telomere length (Fig. 5b). This data strongly suggests that Pof8 contributes to the assembly of Trt1-TER1 complex, independently of its role in ensuring TER1 RNA accumulation in cells.

Likewise, are the TER1 RNA levels affected in ccq1, trt1 or est1 deletion strains (Fig. 3a)?

We did not observe statistically significant changes in *TER1* expression in *ccq1* Δ , *trt1* Δ or *est1* Δ , compared to wild-type cells (Supplementary Fig. 4).

4. The authors use the reduction of *Ccq1* association with telomeres in the *pof8* deletion strain (Fig. 6a) to justify experiments examining telomeric gene silencing (Fig. 6b). It is puzzling, then, that the authors do not include a *ccq1* deletion strain in the transcriptional silencing assays, nor in the assays measuring non-coding RNA abundance (e.g. TERRA). Do the *Pof8* mutants that affect RNA abundance and association of *Pof8* with the *TER1* RNA also affect the association of *Ccq1* with telomeres?

We have added the requested experiments for *ccq1* Δ cells. As previously reported in multiple publications, *ccq1* Δ reduced transcriptional silencing of the *telo::ura4⁺* reporter (Fig. 7b). In addition, we now show that *ccq1* Δ cells indeed show induced expression of TERRA at telomeres (Fig. 7d).

Furthermore, we found that *pof8* xRRM domain mutants do not show reduced *Ccq1* accumulation at telomeres, unlike *pof8* Δ cells, and is consistent with our finding that xRRM mutants retain their ability to repress TERRA expression. In transcriptional silencing assay, xRRM mutants showed partial and less severe disruption in silencing compared to *pof8* Δ .

5. The authors state that "... *Pof8* and *Trt1*TERT appear to mutually promote interaction with *TER1* RNA to allow more efficient assembly of the *TER1*-*Trt1*TERT-*Pof8* complex, analogous to the telomerase "catalytic core" complex of *TER*-*TERT*-*p65* found in *Tetrahymena* cells." However, I don't think that any of the experiments shown prove that *Pof8* is found in a ternary complex with *TER1* and *Trt1*. Does immunoprecipitation of *Pof8* pull down catalytically active telomerase?

We have now included a series of co-IP experiments, which hopefully address the concerns of Reviewer #2. We now provide data that show *Trt1* can co-IP *Pof8* and *Trt1*-*Pof8* interaction is sensitive to RNase-treatment, suggesting that *Trt1* and *Pof8* interaction is mediated by *TER1* RNA (Fig. 3d). We have also shown that co-IP for *Lsm3*-*Pof8* is eliminated in the *pof8*- Δ [289-402] mutant that failed to interact with *TER1*, and in *ter1* Δ cells (Fig. 4d). Finally, we also show that *Trt1*-*Lsm3* interaction is eliminated in *pof8* Δ (Fig. 4e). These additional data, along with our previously included *TER1* co-IP data now strongly support the notion that *Pof8* contributes to a functional telomerase complex, which is mediated by *TER1* RNA (Fig. 4g).

6. The authors state that deletion of *pof8* "completely reversed" (line 101) the telomere elongation observed in mutant strains. However, this doesn't appear to be true for the *Taz1* mutant. What is the discrete band at about 1.5 kb that increases in intensity upon *Pof8* deletion? Is that pattern reproducible?

The 1.5 kb band can often be seen even in wt cells and other strains unrelated to this study at various intensities, and it might represent changes/variations in sub-telomere restriction fragments that hybridize with the telomere probe among different strains. It should also be noted that *taz1* Δ cells already show a more prominent ~1.5 kb band (Fig1b). Furthermore, when we analyzed several additional independent *pof8* Δ *taz1* Δ strains, we found that the 1.5kb band was present in most but not all strains at various intensities, with occasional slight variations in size. Taken together we do not think that this band is relevant to the interpretation of our data, which clearly demonstrates that *pof8* Δ *taz1* Δ cells show dramatic reduction in both intensity and length of telomeres compared to *taz1* Δ cells.

Minor comments:

Figure 5b lacks an untagged negative control lane.

Original Figure 5b panel has been removed, and we now show expression level of various *Pof8* mutants and no tag control lane in new Figure 6d.

Sentence on lines 83 to 85 requires additional references.

We have added additional references.

Reviewers' comments:

Reviewer #1 (Remarks to the Author):

The authors have addressed my concerns. Thank you.

Reviewer #2 (Remarks to the Author):

The authors have responded to the reviewers' comments and provided additional information that strengthens the manuscript. My only remaining concern is about the statistical analysis provided in the supplementary dataset. It is laudable that the authors have included the primary data and the resulting analysis. However, the analysis does not account for multiple comparisons (i.e. the Student's T-test should not be used to make multiple comparisons within the same group of samples); more appropriate would be an ANOVA with a post-hoc test that corrects for multiple comparisons. I am confused about the nature of the "two-tiered" T-test in the Materials and Methods. Is this a "two-tailed" T test?

Reviewer #2 (Remarks to the Author):

The authors have responded to the reviewers' comments and provided additional information that strengthens the manuscript. My only remaining concern is about the statistical analysis provided in the supplementary dataset. It is laudable that the authors have included the primary data and the resulting analysis. However, the analysis does not account for multiple comparisons (i.e. the Student's T-test should not be used to make multiple comparisons within the same group of samples); more appropriate would be an ANOVA with a post-hoc test that corrects for multiple comparisons. I am confused about the nature of the "two-tiered" T-test in the Materials and Methods. Is this a "two-tailed" T test?

We thank Reviewer #2 for noticing an error. Indeed, we meant to state "two-tailed" student t-test. It has been corrected. In addition, we have now included results of one-way Anova analysis, and performed post-hoc analysis by Benjamini *et al.* that control for false discovery rate. We have kept pairwise student t-test as it seems there is no good consensus on how to correct for multiple comparison. Since we report individual data values in Supplementary Dataset, readers can also perform additional statistical analysis.